

# Random matrix universality in dynamical correlation functions at late times

Oscar Bouverot-Dupuis[1,2], Silvia Pappalardi[3], Jorge Kurchan[4],
Anatoli Polkovnikov[5] and Laura Foini[1]*

**1** IPhT, CNRS, CEA, Université Paris Saclay, 91191 Gif-sur-Yvette, France
**2** Université Paris Saclay, CNRS, LPTMS, 91405, Orsay, France
**3** Institut für Theoretische Physik, Universität zu Köln,
Zülpicher Straße 77, 50937 Köln, Germany
**4** Laboratoire de Physique de l'École Normale Supérieure, ENS, Université PSL,
CNRS, Sorbonne Université, Université de Paris, F-75005 Paris, France
**5** Department of Physics, Boston University, Boston, Massachusetts 02215, USA

* laura.foini@ipht.fr

## Abstract

We study the behaviour of two-time correlation functions at late times for finite system sizes considering observables whose (one-point) average value does not depend on energy. In the long time limit, we show that such correlation functions display a ramp and a plateau determined by the correlations of energy levels, similar to what is already known for the spectral form factor. The plateau value is determined, in absence of degenerate energy levels, by the fluctuations of diagonal matrix elements, which highlights differences between different symmetry classes. We show this behaviour analytically by employing results from Random Matrix Theory and the Eigenstate Thermalisation Hypothesis, and numerically by exact diagonalization in the toy example of a Hamiltonian drawn from a Random Matrix ensemble and in a more realistic example of disordered spin glasses at high temperature. Importantly, correlation functions in the ramp regime do not show self-averaging behaviour, and, at difference with the spectral form factor the time average does not coincide with the ensemble average.

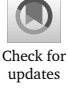
# 1  Introduction

Universal properties derived from Random Matrix Theory (RMT) [1] have demonstrated exceptional robustness in describing quantum non-integrable systems [2]. Recently, there has been a growing interest in applying these concepts to systems with many interacting degrees of freedom, with developments from various fields, including the characterisation of chaotic behaviour in quantum many-body dynamics [3–6] and the emergence of gravity in high-energy physics [7–10].

RMT universality has long been observed in the correlations that govern close by *energy levels* in "non-integrable" Hamiltonians, at least at high enough energies. The paradigmatic observation generically made for non-integrable models is the phenomenon of level repulsion which forbids excessively small energy gaps and enforces some spectral rigidity. Correlations between energy levels are accurately diagnosed by their Fourier transform called the Spectral Form Factor (SFF). At short times, the SFF always decays during the *slope* regime, which usually arises from the disconnected part of the pair distribution of energy levels. At late times, the behaviour of the SFF depends on the nature of the system considered. For chaotic or RMT systems, the SFF first shows a clear growth, usually dubbed the *ramp*, which is a fingerprint of level repulsion manifested in the connected pair density of levels. At even larger times comparable with the Hilbert space dimension (the Heisenberg time), the SFF reaches a *plateau* and fluctuates around this constant value. For integrable models, the ramp is absent, and the slope instead directly transitions to a plateau. In RMT and chaotic quantum many-body systems, the averaged (over an ensemble or time) SFF can thus be decomposed in two terms

$$\overline{\mathrm{SFF}}(t) = \mathrm{SFF}_{\mathrm{d}}(t) + \mathrm{SFF}_{\mathrm{c}}(t), \tag{1}$$

where the first *disconnected* term usually describes the slope and the second *connected* term generically contains the universal late-time behaviour predicted by RMT.

RMT universality has also been discussed in the context of chaotic *eigenvectors*, extending the pioneering idea that eigenfunctions of the Hamiltonian can be modelled as random vectors [11]. While the eigenvectors of rotationally-invariant random matrix ensembles are structureless, and therefore too simple to account for some form of locality present in more realistic Hamiltonians, many of their properties can be generalised in order to take into account some energy dependence. This has been done within the Eigenstate Thermalisation Hypothesis (ETH), first in its original version [3,12–14] and later in more general extensions [15,16]. Although the ETH's original scope was to describe the onset of thermal equilibrium [13,14,17], the objects of interest described within the ETH are dynamical correlation functions, typically seen in the thermodynamic limit. While the ETH remains formally unproven, a good working

hypothesis is that it holds for systems exhibiting level repulsion (an exception being made for systems with quantum many-body scars, which display a few "athermal" eigenstates [18–20]).

In this work, we put together the universal properties of energy levels and those of the ETH eigenvectors to characterise the late-time behaviour of two-time correlation functions (at equilibrium) in large but finite systems. We show that, upon averaging and for well-chosen observables, correlation functions of some observables display a ramp and a plateau similar to that of the SFF. This leads to the decomposition

$$\overline{\Gamma}(t) = \Gamma_{\mathrm{d}}(t) + \Gamma_{\mathrm{c}}(t), \tag{2}$$

for a suitably-defined connected correlation function $\Gamma(t)$ which depends on the RMT universality class of the system. We show that $\Gamma_{\mathrm{d}}(t)$ amounts for the non-universal physical dynamical correlations (i.e. the Fourier transform of the smooth ETH off-diagonal function), while $\Gamma_{\mathrm{c}}(t)$ can be written as the convolution of the latter with the spectral correlations, and thus represents the spectral correlations responsible for the late-time ramp-plateau behaviour.

Similar observations have been drawn in the high-energy literature [7, 21–24], random circuits [25] or hydrodynamics [26]. While the mechanism by which such a ramp can appear is easy to grasp, a justification of when it can be actually observed is instead rather subtle. Here, we provide a careful analytical study using the ETH approach. Our main assumption is that the average over energy levels and eigenvectors decouples. This is a standard result for rotationally-invariant Hamiltonians, and we here assume it also holds true for realistic many-body systems satisfying the ETH. We also show numerical evidence for our predictions in macroscopic many-body systems. While we chose to study numerically spin glass models, the same conclusions could have been drawn considering the SYK model, which however was not convenient for us as we need to treat separately different RMT universality classes. A key requirement to expect such a RMT behaviour is to consider observables whose expectation values (one-point functions) do not show any energy dependence. This allows, in absence of degenerate energy levels, the plateau to arise mainly from the fluctuations of the observable's diagonal matrix elements, making it exponentially small in the system size. This is to be put in contrast with the case considered in [27,28], and recently in [29], where an explicit energy dependence leads to a plateau which scales polynomially with the system size. Besides this in our calculations we assumed that the ETH function $f(\omega)$ which characterises the off-diagonal matrix elements has a finite value for $\omega \to 0$.

Interestingly enough, at late times when dynamical correlation functions present a ramp, their fluctuations are so large that *their time-average does not reproduce their ensemble-average*, contrary to the SFF. This is at stark contrast with usual quantities that are generically computed within the ETH and have a smooth self-averaging behaviour. The lack of self-averaging means that working with a fictitious ensemble as is usually assumed in the ETH is tricky. In order to show our predictions we preferred to work with disordered systems where we unequivocally specify the ensemble of Hamiltonians as that generated by all instances of disorder.

We organise the manuscript by presenting first the results from RMT in Sec. 2, where the Hamiltonian is drawn from the GOE or the GUE. In order to see the ramp, it appears important to distinguish between these two ensembles. Section 3 then presents the results for the many-body problems, justifying our predictions with ETH and RMT arguments, and then studying numerically two examples of spin glasses at infinite temperature exhibiting GOE or GUE statistics.

## 2 Random matrix Hamiltonians

### 2.1 Two-time correlations and SFF in RMT

In this section, we review the simpler instance in which the system's Hamiltonian $H$ is drawn from a rotationally-invariant ensemble [1], where Eq.(2) acquires a particularly straightforward form. For simplicity, in the numerical simulation, we will specialise in a $\mathcal{N} \times \mathcal{N}$ Gaussian ensemble. Specifically, we consider the Orthogonal and Unitary Gaussian ensembles (GOE and GUE), defined over symmetric (and Hermitian) matrices, described by the probability distribution

$$P(H) \propto \exp\left(-\beta_{\mathrm{RMT}} \frac{\mathcal{N}}{4} \mathrm{Tr}(H^2)\right), \tag{3}$$

where $\beta_{\mathrm{RMT}} = 1$ for the GOE and $\beta_{\mathrm{RMT}} = 2$ for the GUE [1, 30]. The eigenvalues $\{E_i\}$ density of states is denoted

$$\rho(E) = \sum_i \delta(E - E_i), \tag{4}$$

and, for large $\mathcal{N}$, the associated single-eigenvalue distribution $\rho(E)/\mathcal{N}$ converges to the Wigner semicircle distribution $p(E) = \frac{1}{2\pi}\sqrt{4 - E^2}$ [31]. Let us stress again that while here, for simplicity, we focus on the GOE or GUE ensemble, the results could be generalised to any rotationally-invariant ensemble [1], suitably changing the asymptotic spectral function. This will only change the slope part.

Eigenvalues correlations are encoded in the Spectral Form Factor, which, for a fixed Hamiltonian, is defined as

$$\mathrm{SFF}(t) = \left|\langle e^{-iHt}\rangle\right|^2, \tag{5}$$

where $\langle \bullet \rangle = \mathrm{Tr}(\bullet)/\mathcal{N}$ is the infinite-temperature canonical average. In the case of random matrix models, the SFF displays well-known features and leads to the so-called *slope-dip-ramp-plateau* picture [7, 31, 32]. Let us recall here its basic properties. We will denote ensemble averages by $\overline{\bullet}$. In continuous energy variables, the ensemble average of the SFF reads

$$\overline{\mathrm{SFF}}(t) = \frac{1}{\mathcal{N}^2} \int \mathrm{d}E_1 \, \mathrm{d}E_2 \, \overline{\rho(E_1)\rho(E_2)} e^{i(E_1 - E_2)t}, \tag{6}$$

and can be split into two parts $\overline{\mathrm{SFF}}(t) = \mathrm{SFF}_{\mathrm{d}}(t) + \mathrm{SFF}_{\mathrm{c}}(t)$, as in Eq.(1). The disconnected part, denoted $\mathrm{SFF}_{\mathrm{d}}$, is defined as

$$\mathrm{SFF}_{\mathrm{d}}(t) = \frac{1}{\mathcal{N}^2} \int \mathrm{d}E_1 \, \mathrm{d}E_2 \, \overline{\rho(E_1)} \, \overline{\rho(E_2)} e^{i(E_1 - E_2)t} = \left|\frac{1}{\mathcal{N}} \int \mathrm{d}E \, e^{iEt} \overline{\rho(E)}\right|^2, \tag{7}$$

which encodes for the average $\rho(E)$ and hence accounts for the non-universal early-time decay. The connected part, $\mathrm{SFF}_{\mathrm{c}}$, encodes the level density correlations, and it accounts for the universal RMT level repulsion as

$$\mathrm{SFF}_{\mathrm{c}}(t) = \overline{\mathrm{SFF}}(t) - \mathrm{SFF}_{\mathrm{d}}(t) = \frac{1}{\mathcal{N}^2} \int \mathrm{d}E_1 \, \mathrm{d}E_2 \left(\overline{\rho(E_1)\rho(E_2)} - \overline{\rho(E_1)} \, \overline{\rho(E_2)}\right) e^{i(E_1 - E_2)t}. \tag{8}$$

At early times, the SFF is dominated by the decay (i.e. the *slope*) of $\mathrm{SFF}_{\mathrm{d}}$. In the case of Gaussian ensembles, the slope is given by

$$\mathrm{SFF}_{\mathrm{d}}(t) = \left(\frac{J_1(2t)}{t}\right)^2 \sim \frac{1}{t^3}, \tag{9}$$

with $J_1$ the 1$^{\mathrm{st}}$ Bessel function of the first kind. Around the time $t_{\mathrm{dip}} \sim \sqrt{\mathcal{N}}$, the SFF stops decreasing (*dip*) and the contribution $\mathrm{SFF}_{\mathrm{c}}$ becomes dominant. It grows (*ramp*) before saturating

at a constant value (*plateau*). In RMT, the connected two-point correlations of the density of states can be expressed as

$$\overline{\rho(E_1)\rho(E_2)} - \overline{\rho(E_1)}\,\overline{\rho(E_2)} = \mathcal{N}^2 \mathcal{R}[\mathcal{N}(E_1 - E_2)], \tag{10}$$

where the exact expressions of $\mathcal{R}$ for the different Gaussian ensembles can be found in Ref. [31]. For the GUE, taking the Fourier transform of $\mathcal{R}$ yields a strictly linear ramp

$$\text{SFF}_\text{c}(t) = \begin{cases} \frac{t}{2\mathcal{N}^2}, & \text{for } t < 2\mathcal{N}, \\ \frac{1}{\mathcal{N}}, & \text{for } t > 2\mathcal{N}, \end{cases} \tag{11}$$

while, for the GOE, logarithmic corrections appear

$$\text{SFF}_\text{c}(t) = \begin{cases} \frac{t}{\mathcal{N}^2} - \frac{t}{2\mathcal{N}^2}\ln\left(1 + \frac{t}{\mathcal{N}}\right), & \text{for } t < 2\mathcal{N}, \\ \frac{2}{\mathcal{N}} - \frac{t}{2\mathcal{N}^2}\ln\left(\frac{t+L}{t-L}\right), & \text{for } t > 2\mathcal{N}. \end{cases} \tag{12}$$

It is worth noting that the plateau value is always $\mathcal{N}^{-1}$, while the dip value is $t_\text{dip}/\mathcal{N}^2 \sim \mathcal{N}^{-3/2}$.

Let us now discuss the dynamical correlations of a local observable $A$ and consider the connected two-point correlation function

$$C(t) = \frac{1}{2}\langle\{A(t), A(0)\}\rangle - \langle A \rangle^2, \tag{13}$$

where $\{\bullet, \bullet\}$ is the anticommutator. Its ensemble average $\overline{C}(t)$ can be rewritten in the Hamiltonian's basis $\{|E_i\rangle, E_i\}_i$ as

$$\overline{C}(t) = \frac{1}{\mathcal{N}}\sum_{i,j}\overline{|A_{ij}|^2 e^{i(E_i - E_j)t}} - \overline{\langle A \rangle^2} = \frac{1}{\mathcal{N}}\sum_{i\neq j}\overline{|A_{ij}|^2 e^{i(E_i - E_j)t}} + \frac{1}{\mathcal{N}}\sum_i \overline{A_{ii}^2} - \langle A \rangle^2, \tag{14}$$

with $A_{ij} = \langle E_i|A|E_j\rangle$ and where in the last equality, we have removed the ensemble average over $\langle A \rangle$ since the trace does not depend on the basis it is computed in. In rotationally-invariant ensembles as the GOE or the GUE, the probability distribution of eigenvectors and eigenvalues factorises, so that we can take their averages separately. For $\mathcal{N} \gg 1$, the averages over the matrix elements can be computed by using (see Ref. [33] for a proof of high-order expectations)

$$\overline{|A_{ij}|^2} = \begin{cases} \langle A \rangle^2 + \frac{1}{\mathcal{N}}\frac{2\kappa_2}{\beta_\text{RMT}}, & \text{if } i = j, \\ \frac{1}{\mathcal{N}}\kappa_2, & \text{if } i \neq j, \end{cases} \tag{15}$$

where we have introduced the second free cumulant $\kappa_2 = \langle|A|^2\rangle - \langle A \rangle^2$. This leads to

$$\overline{C}(t) \overset{\mathcal{N}\gg 1}{\simeq} \frac{\kappa_2}{\mathcal{N}^2}\overline{\sum_{i\neq j}e^{i(E_i - E_j)t}} + \frac{2\kappa_2}{\mathcal{N}\beta_\text{RMT}} = \kappa_2\overline{\text{SFF}}(t) + \left(\frac{2}{\beta_\text{RMT}} - 1\right)\frac{\kappa_2}{\mathcal{N}}, \tag{16}$$

with the spectral form factor (SFF) defined as $\text{SFF}(t) = \frac{1}{\mathcal{N}^2}\sum_{i,j}e^{i(E_i - E_j)t}$. It is natural to introduce the shifted connected correlation function $\Gamma$ such that

$$\Gamma(t) = C(t) - \left(1 - \frac{\beta_\text{RMT}}{2}\right)\lim_{t\to\infty}\overline{C}(t), \tag{17}$$

where in the above, the infinite time limit of $\overline{C}(t)$ is well defined, i.e. $\lim_{t\to\infty}\overline{C}(t) = \frac{2}{\beta_\text{RMT}}\frac{\kappa_2}{\mathcal{N}}$. With this notation, one has

$$\overline{\Gamma}(t) = \kappa_2\overline{\text{SFF}}(t). \tag{18}$$

Note that the shift that we perform is necessary to observe the ramp, similarly to what happens in the partial spectral form factor [34]. This identity shows that, for a Hamiltonian drawn from random ensembles, the behaviour of a two-point function can be extracted from that of the SFF [7,31]. The present analysis emphasises the importance of the symmetry classes for extracting the eigenvalue correlations from dynamical correlators, which is encoded in the re-scaled Eq.(17) and not in the bare connected correlator. From Eq.(18), it follows that, for random matrices, the decomposition of dynamical correlations in Eq.(2) holds upon identifying $\Gamma_d(t) = \kappa_2 \text{SFF}_d(t)$ and $\Gamma_c(t) = \kappa_2 \text{SFF}_c(t)$.

## 2.2 Lack of self-averaging

While the previous calculation characterised the ensemble-averaged quantity $\overline{\Gamma}(t)$, this section aims at estimating the single instance fluctuations $\delta\Gamma(t) = \Gamma(t) - \overline{\Gamma}(t)$. In random matrices, because $\overline{|A_{ij}|^4} \propto \overline{|A_{ij}|^2}^2$ with a proportionality constant depending on the ensemble, we expect that matrix elements behave as

$$|A_{ij}|^2 = \frac{1}{\mathcal{N}}\kappa_2 + \frac{c}{\mathcal{N}}\xi_{ij}, \quad \text{for } i \neq j, \tag{19}$$

with $\{\xi_{ij}\}_{ij}$ a set of weakly correlated random variables with mean zero, variance one, and whose exact distributions depend on the observable considered. In a single instance $\Gamma(t)$, summing the $\mathcal{N}^2$ noise contributions gives rise to a noisy component of strength $\mathcal{N}^{-1}$, i.e.

$$\Gamma(t) = \overline{\Gamma}(t) + \mathcal{O}(\mathcal{N}^{-1}). \tag{20}$$

While this does not significantly obstruct the recovery of $\overline{\Gamma}(t)$ in the slope of order $\mathcal{O}(1)$ and the plateau of order $\mathcal{O}(\mathcal{N}^{-1})$, fluctuations completely hide the signal close to the dip which scales as $\mathcal{N}^{-3/2}$ (see Sec. 2.1).

This is confirmed by the numerics in the Gaussian ensemble case (see the right panels of Figs. 1-2 below) where we plot the single-sample $\Gamma(t)$. Contrary to the Spectral Form Factor, where the small-window time average reproduces the dip-ramp behaviour, for the dynamical correlation function $\Gamma(t)$, the dip and ramp are nowhere to be seen even after averaging over a small time window. The fact that, in the RMT toy models, the noise $\delta\Gamma(t)$ can become larger than the average signal $\overline{\Gamma}(t)$ thus results in correlation functions not being self-averaging, and in the small-window time-average not reproducing the ensemble-average in contrast to what is usually expected in ETH in the decaying part.

## 2.3 Numerical analysis for RMT ensembles

We here check numerically the previous findings in the case of the gaussian ensembles sampled with Eq. (3).

### 2.3.1 The Gaussian orthogonal ensemble

We consider here a Hamiltonian from the GOE, i.e. drawn from Eq. (3) with $\beta_{\text{RMT}} = 1$, and take the observable $A = S_j^z$, with $j$ an arbitrary site index. Fig. 1 (right) shows the behaviour of $\Gamma(t) = C(t) - \kappa_2/\mathcal{N}$ and of the SFF for a single realisation of the GOE Hamiltonian. Upon averaging over some small time window, one cannot resolve the ramp in $\Gamma(t)$. However, if one considers the average $\overline{\Gamma}(t)$ over many realisations of the GOE Hamiltonian, the slope-dip-ramp-plateau becomes clear as shown in Fig. 1 (left). This is in contrast with the SFF where, despite the fluctuations, the ramp is visible also in a single instance, and the time average of one single instance coincides with the ensemble average.

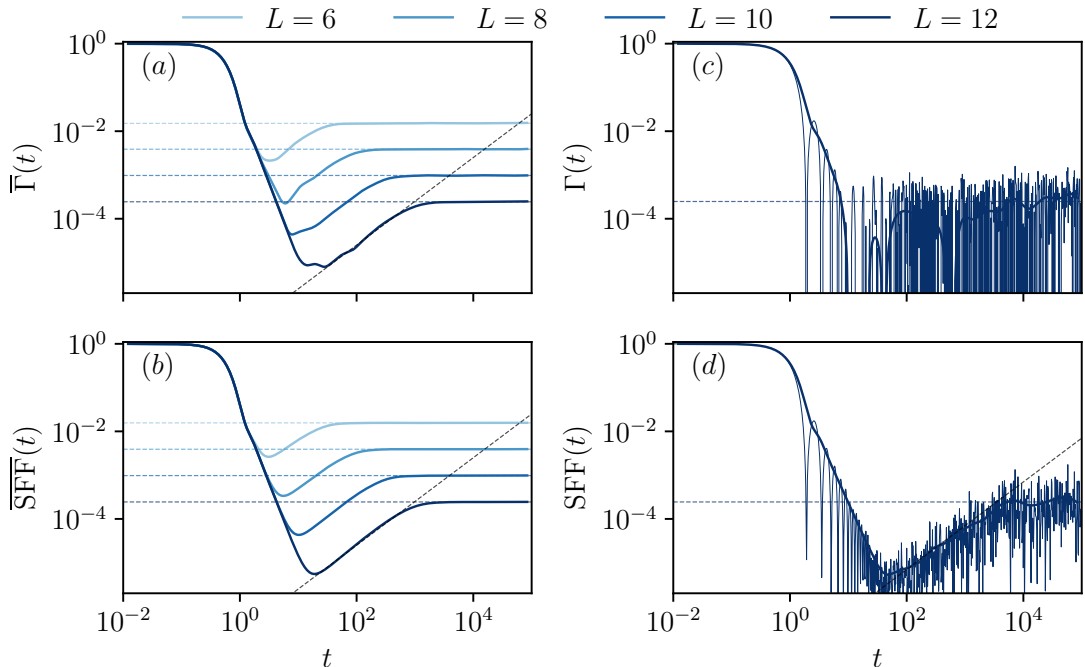

Figure 1: Left: (a) Shifted correlation function $\overline{\Gamma}(t)$ and (b) spectral form factor $\overline{\mathrm{SFF}}(t)$ averaged over 1000 GOE Hamiltonians. Right: (c) Shifted correlation function $\Gamma(t)$ and (d) spectral form factor $\mathrm{SFF}(t)$ obtained from a single GOE Hamiltonian. Plots (c) and (d) are for $L = 12$ spins. In all plots, the thin curves are the exact result while the thick curves are a smoothed version obtained by convoluting with a Gaussian kernel. The horizontal dashed lines are the expected plateau values coming from the diagonal ensemble. The increasing oblique dashed lines are fits $\propto t$.

### 2.3.2 The Gaussian unitary ensemble

We now consider a Hamiltonian from the GUE (Eq. (3) with $\beta_{\mathrm{RMT}} = 2$). Fig. 2 (right) shows the behaviour of $\Gamma = C(t)$ and of the SFF for a single realisation of the GUE Hamiltonian. Upon averaging over some small time window, the slope-dip-ramp-plateau feature becomes visible in the SFF, while the ramp cannot be resolved in $\Gamma$. If one averages $\Gamma$ and the SFF over many realisations of the GUE Hamiltonian, the ramp-plateau becomes much more clear as shown in Fig. 2 (left).

### 2.3.3 Numerical evaluation of the fluctuations

In order to quantify the fluctuations of correlations relative to their average in Eq.(20), we consider the variance of $\Gamma(t)$ normalised by its average, namely

$$\overline{\delta\Gamma^2}(t)/\overline{\Gamma}^2(t), \tag{21}$$

which we compare with the same quantity involving the SFF

$$\overline{\delta\mathrm{SFF}^2}(t)/\overline{\mathrm{SFF}}^2(t), \tag{22}$$

where $\delta\mathrm{SFF}(t) = \mathrm{SFF}(t) - \overline{\mathrm{SFF}}(t)$ are the fluctuations of the SFF. The results for the GOE case are summarised in Fig. 3, and a similar picture can be drawn for the GUE case. While the SFF has a bounded noise-to-signal ratio, for the correlation function $\Gamma$, this ratio increases with the system size $L$ in the dip and ramp, signalling there that the fluctuations $\delta\Gamma(t)$ completely dwarf the ensemble average $\overline{\Gamma}(t)$.

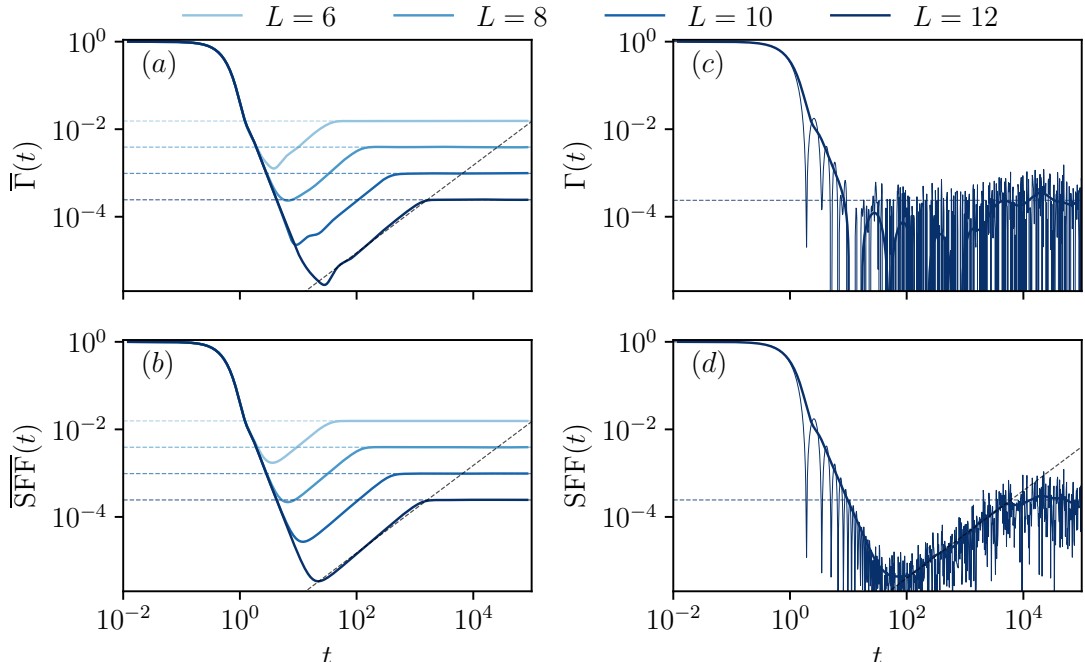

Figure 2: Left: $(a)$ Shifted correlation function $\overline{\Gamma}(t)$ and $(b)$ spectral form factor $\overline{\text{SFF}}(t)$ averaged over 1000 GUE Hamiltonians. Right: $(c)$ Shifted correlation function $\Gamma(t)$ and $(d)$ spectral form factor $\text{SFF}(t)$ obtained from a single GUE Hamiltonian. Plots $(c)$ and $(d)$ are for $L = 12$ spins. In all plots, the thin curves are the exact result while the thick curves are a smoothed version obtained by convoluting with a Gaussian kernel. The horizontal dashed lines are the expected plateau values coming from the diagonal ensemble. The increasing oblique dashed lines are fits $\propto t$.

This underscores a crucial difference between the SFF and dynamical correlations, even in models of random matrices. Unlike the SFF, dynamical correlations do not self-average when it comes to time averages over small intervals. This is due to fluctuations in the matrix elements and, therefore, in the eigenvectors.

## 3 Many-body Hamiltonian

In this section, we discuss how the previous results can be extended to generic many-body interacting Hamiltonians of $L$ constituents that obey the ETH. We show that the ensemble average of the properly-defined two-point functions can be decomposed as

$$\overline{\Gamma}(t) = \Gamma_{\text{d}}(t) + \Gamma_{\text{c}}(t), \tag{23}$$

where in the large $L$ limit, the first contribution, related to the disconnected spectral correlations, encodes all the physical dynamical two-point function as

$$\Gamma_{\text{d}}(t) = \int d\omega \, |f_{e_0}(\omega)|^2 e^{i\omega t}, \tag{24}$$

where $f_{e_0}(\omega)$ is the smooth function appearing in the off-diagonal ETH ansatz (see below). On the other hand, the second contribution $\Gamma_{\text{c}}$ is related to the connected spectral correlations

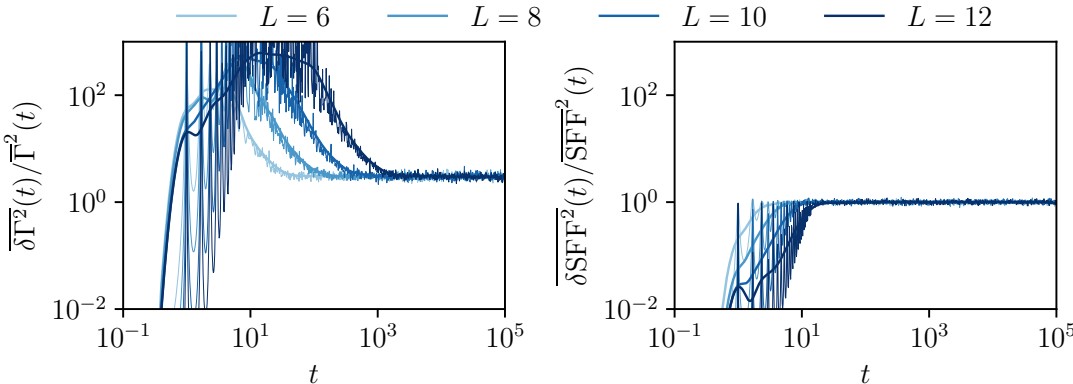

Figure 3: Normalised fluctuations of the shifted correlation function $\Gamma(t)$ and the spectral form factor $\text{SFF}(t)$ averaged over 1000 GOE Hamiltonians at inverse temperature $\beta = 0$. The thin curves are the exact result while the thick curves are a smoothed version obtained by convoluting with a Gaussian kernel.

and generically features a late-time ramp followed by a plateau as for the SFF since

$$\Gamma_{\text{c}}(t) \sim \int \mathrm{d}t' \, C_{\text{d}}(t - t')\text{SFF}_{\text{c}}(t') \simeq f_{e_0}^2(0)\text{SFF}_{\text{c}}(t), \tag{25}$$

where we assumed that the function $f_e(\omega)$ has a finite limit for $\omega \to 0$.

## 3.1 Two-time correlations and SFF within ETH

For definiteness, we focus on infinite temperature and consider 2-level systems, such that the total Hilbert space dimension is $\mathcal{N} = \dim \mathcal{H} = 2^L$. Given a generic Hamiltonian $\hat{H}$ with energy levels $\{E_i\}_i$, we define the density of states $\rho(E)$ of a many-body system as in Eq.(4). It is related to the thermodynamic entropy $S(E)$ as

$$\overline{\rho(E)} \simeq e^{S(E)}, \tag{26}$$

where $\overline{\bullet}$ now denotes some form of averaging over an ensemble to be specified. It could be some small perturbation of the system, the disorder in a random Hamiltonian or some energy/time window. In general, one can introduce a thermodynamically well-defined energy density $e = E/L$ and entropy density $s(e = E/L) = S(E)/L$ so the density of states is exponentially large in the number $L$ of degrees of freedom (and the level spacing exponentially small). We will assume that no degeneracies are present in the spectrum.

In the following, we aim to characterise dynamical correlations and thus assume the ETH ansatz for the matrix elements of a local observable $A$. The off-diagonal matrix elements are expected to fluctuate as [14]

$$\overline{|A_{ij}|^2} \simeq \overline{\rho(E_{ij}^+)}^{-1} f_{e_{ij}^+}^2(\omega_{ij}) \qquad (i \neq j), \tag{27}$$

where $e_{ij}^+ = E_{ij}^+/L = (E_i + E_j)/(2L)$ is the mean intensive energy and $\omega_{ij} = E_i - E_j$ is the energy difference. The diagonal matrix elements are characterised by a microcanonical average $\mathcal{A}(e)$ of order $\mathcal{O}(1)$ and small fluctuations which lead to

$$\overline{A_{ii}^2} \simeq \mathcal{A}^2(e_i) + \overline{\delta A_{ii}^2}. \tag{28}$$

The fluctuations of diagonal elements are related to those of off-diagonal elements due to the local rotational invariance of the many-body Hamiltonian (see for instance [35]), implying that

$$\overline{\delta A_{ii}^2} \simeq \frac{2}{\beta_{\text{RMT}}} \overline{\rho(E_i)}^{-1} f_{e_i}^2(\omega_{ii}=0), \tag{29}$$

with $\beta_{\text{RMT}} = 1$ for real Hamiltonians (GOE-like) and $\beta_{\text{RMT}} = 2$ for complex Hamiltonians (GUE-like).

An important assumption made in the following is that *the observable A has an average $\mathcal{A}(e)$ that does not depend on e, while A still fluctuates* as in Eq. (29). In fact, if $\partial_e \mathcal{A}(e)$, with $e = E/L$ the energy density, is of order one in the system's size, there will always be polynomial corrections in $L$ to the two-point function coming from the diagonal part of the matrix elements after integration by saddle-point (see below). This hypothesis is required to avoid extra terms which could hide the slope-dip-ramp-plateau behaviour we are looking for [29], because of polynomially small/large corrections to the plateau, which may be understood by the fact that the fluctuations of the observable are then driven by the large local excursions of energy, see e.g. Ref. [29]. The assumption of no energy dependency is, for instance, guaranteed in the presence of symmetries and conservation laws. In particular one can study a local observable protected by some global conservation law in a disordered system, as will do in the following.[1] Other examples could be drawn from disordered systems in their paramagnetic phase and at high energies where the ensemble average due to disorder implies the vanishing of expectation values of several quantities, at least in a certain range of temperatures (for instance the (mixed) $p$-spin model in transverse field), or from Floquet systems.

Drawing inspiration from the previous section on RMT, we now wish to establish a link between the large-time behaviour of connected correlation functions defined in Eq. (13) and the SFF. To simplify the analysis, we consider the system at infinite temperature ($\beta = 0$), since the extension to finite $\beta$ is a natural generalization. In fact, even at finite temperatures, energy-independent observables must still be considered to observe the exponentially small ramp, ensuring a simple generalization.

In the limit of large, yet finite, system size, the ensemble-averaged correlation function reads

$$\overline{C}(t) = \frac{1}{2Z} \int dE_1 \, dE_2 \, \overline{\rho(E_1)\rho(E_2)} \, \overline{|A_{E_1,E_2}|^2} e^{i(E_2-E_1)t} + c.c. - \langle A \rangle^2. \tag{30}$$

As for the rotationally-invariant ensembles considered in Sec. 2.1, the equation above assumes that the probability distribution over eigenvectors and eigenvalues of the Hamiltonian factorises. Next, distinguishing between off-diagonal and diagonal elements of $A$ and using the ETH ansatz of Eqs. (27-29) leads to

$$\overline{C}(t) = \frac{1}{Z} \int_{|\omega| \gtrsim \overline{\rho(E)}^{-1}} dE \, d\omega \, \frac{\overline{\rho(E+\omega/2)\rho(E-\omega/2)}}{\overline{\rho(E)}} f_e^2(\omega) e^{i\omega t}$$
$$+ \frac{1}{Z} \int dE \, \frac{\overline{\rho(E)^2}}{\overline{\rho(E)}} \left[ \mathcal{A}^2(e) + \frac{2}{\beta_{\text{RMT}}} \frac{f_e^2(0)}{\overline{\rho(E)}} \right] - \langle A \rangle^2, \tag{31}$$

---

[1]To be precise, we will focus on the following mechanism. We consider a disordered spin system that conserves the total magnetization $M^z = \sum_i \sigma_i^z$ and restricts ourselves to one of the sectors of the Hilbert space with fixed total magnetization $M \in \mathbb{Z}$. In the restricted Hilbert space, the thermal average of the total magnetization is, of course, $\langle M^z \rangle_\beta = M$. Then, we consider a given site $i$ and its local magnetisation $\langle \sigma_i^z \rangle_\beta$ which fluctuates depending on the disorder realisation. However, upon ensemble averaging, site-permutation symmetry is restored and $\overline{\langle \sigma_i^z \rangle_\beta} = M/L$. Therefore this quantity does not depend on temperature and energy.

where $E = (E_1 + E_2)/2$ and $\omega = E_1 - E_2$. Due to the standard saddle-point argument of the ETH, the contribution of the canonical average $\langle A \rangle^2$ exactly cancels that of the microcanonical average $\mathcal{A}(e)$ provided that $\partial_e \mathcal{A}(e)$ does not depend on energy on the saddle-point, as discussed above. Shuffling terms between both integrals gives

$$\overline{C}(t) = \frac{1}{Z} \int \mathrm{d}E\, \mathrm{d}\omega \frac{\overline{\rho(E+\omega/2)\rho(E-\omega/2)}}{\overline{\rho(E)}} f_e^2(\omega) e^{i\omega t} + \left( \frac{2}{\beta_{\mathrm{RMT}}} - 1 \right) \frac{1}{Z} \int \mathrm{d}E \frac{\overline{\rho(E)^2}}{\overline{\rho(E)}^2} f_e^2(0), \tag{32}$$

which, as for the RMT case in Eq. (17), suggests to introduce

$$\Gamma(t) = C(t) - \left( 1 - \frac{\beta_{\mathrm{RMT}}}{2} \right) \lim_{t \to \infty} C(t), \tag{33}$$

where $\lim_{t \to \infty} C(t) = \int \mathrm{d}E \frac{\overline{\rho(E)^2}}{\overline{\rho(E)}^2} f_e^2(0)$. We are now in the position to evaluate the ensemble average. Looking at Eq. (32), one is left with evaluating the two-point correlations of the density of states $\overline{\rho(E+\omega/2)\rho(E-\omega/2)}$. This can be done by separating the disconnected correlations, which capture the short-time physics of the many-body system, from the connected correlations, which are expected to display an RMT behaviour. Using Eq. (26), the disconnected part turns out to be

$$\overline{\rho(E+\omega/2)}\,\overline{\rho(E-\omega/2)} \simeq \overline{\rho(E)}^2 e^{\frac{1}{4L} \frac{\mathrm{d}^2 s(e)}{\mathrm{d}e^2} \omega^2}, \tag{34}$$

with $\frac{1}{L} \frac{\mathrm{d}^2 s}{\mathrm{d}e^2} = -\beta^2 / C_V$ and $C_V$ the heat capacity. This gives a contribution $\Gamma_{\mathrm{d}}$ to the shifted correlation function $\Gamma$ defined by

$$\Gamma_{\mathrm{d}}(t) = \frac{1}{Z} \int \mathrm{d}E\, \mathrm{d}\omega\, \overline{\rho(E)} e^{\frac{1}{4L} \frac{\mathrm{d}^2 s(e)}{\mathrm{d}e^2} \omega^2} f_e^2(\omega) e^{i\omega t}. \tag{35}$$

As in the RMT case, $\Gamma_{\mathrm{d}}$ is expected to give rise to the slope since it comes from the disconnected part of the correlations between energy levels. It is furthermore expected to be self-averaging with respect to the ensemble average, as is usual within the ETH. In the large system size limit, the factor $\exp\left( \frac{1}{4L} \frac{\mathrm{d}^2 s(e)}{\mathrm{d}e^2} \omega^2 \right)$ can be dropped and a saddle point approximation (also done for the partition function $Z$) leads to

$$\Gamma_{\mathrm{d}}(t) = \int \mathrm{d}\omega\, f_{e_0}^2(\omega) e^{i\omega t}, \tag{36}$$

where $e_0$ is such that $\frac{\mathrm{d}s}{\mathrm{d}e}(e_0) = 0$. Note that, if the observable were dependent on the energy, this quantity would be corrected by an additional term $\Delta_e^2 (\partial_e \mathcal{A})^2$, with $\Delta_e^2 \simeq N^{-1}$ the energy fluctuations, which would hide the ramp and the plateau, see e.g. Ref. [29].

We now turn to the connected two-point correlations of the density of states. Since this part is expected to have a random matrix behaviour, we generalise Eq. (10) for a many-body system as

$$\overline{\rho(E_1)\rho(E_2)} - \overline{\rho(E_1)}\,\overline{\rho(E_2)} = \pi^2 \overline{\rho(E_1)}\,\overline{\rho(E_2)} \mathcal{R}\big[ \mathcal{N}(\Phi(E_1) - \Phi(E_2)) \big], \tag{37}$$

where the function $\Phi$ is the function relating to spectral unfolding [36] and it is defined as

$$\frac{\mathrm{d}\Phi}{\mathrm{d}E} = \frac{\pi}{\mathcal{N}} \overline{\rho(E)}. \tag{38}$$

The equation is understood as substituting the density of states $\frac{\mathcal{N}}{\pi}$ found in the bulk of the Gaussian ensembles by that of the many-body system, $\overline{\rho(E)}$. This gives rise to the following random matrix contribution

$$
\begin{aligned}
\Gamma_{\mathrm{c}}(t) &= \frac{1}{Z} \int \mathrm{d}E\,\mathrm{d}\omega\,\pi^2 \frac{\overline{\rho(E+\omega/2)}\,\overline{\rho(E-\omega/2)}}{\overline{\rho(E)}} \mathcal{R}\big[\mathcal{N}(\Phi(E+\omega/2)-\Phi(E-\omega/2))\big] f_e^2(\omega)e^{i\omega t} \\
&= \frac{1}{Z} \int \mathrm{d}E\,\mathrm{d}\omega\,\pi^2 \overline{\rho(E)}\mathcal{R}[\pi\overline{\rho(E)}\omega]f_e^2(\omega)e^{i\omega t}\,,
\end{aligned}
\tag{39}
$$

where the last line is a small $\omega$ expansion valid at large time $t$. For large systems, a saddle-point computation yields

$$
\Gamma_{\mathrm{c}}(t) \simeq \pi^2 \int \mathrm{d}\omega\,\mathcal{R}[\pi\overline{\rho(E_0)}\omega]f_{e_0}^2(\omega)e^{i\omega t}\,.
\tag{40}
$$

To understand this expression, one has to compare it with the SFF in many-body systems. Here, we reintroduce the temperature as it is convenient to study the SFF as a function of the complex variable $z = \beta + it$ [37]. The average of the SFF is then given by

$$
\overline{\mathrm{SFF}}(t) = \frac{\overline{|Z(\beta+it)|^2}}{\overline{Z(\beta)}^2} = \mathrm{SFF}_{\mathrm{d}}(t) + \mathrm{SFF}_{\mathrm{c}}(t)\,,
\tag{41}
$$

and, following the steps of the derivation done previously for $\Gamma$, the contributions $\mathrm{SFF}_{\mathrm{d}}$ and $\mathrm{SFF}_{\mathrm{c}}$ are expressed as

$$
\mathrm{SFF}_{\mathrm{d}}(t) = \sqrt{\left|\frac{\mathrm{d}^2 s}{\mathrm{d}e^2}(e_\beta)\right| \frac{1}{4\pi L}} \int \mathrm{d}\omega\, e^{\frac{1}{4L}\frac{\mathrm{d}^2 s(e)}{\mathrm{d}e^2}\omega^2} e^{i\omega t}\,,
\tag{42}
$$

$$
\mathrm{SFF}_{\mathrm{c}}(t) \sim \int \mathrm{d}\omega\,\pi^2 \mathcal{R}[\pi\overline{\rho(E_\beta)}\omega]e^{i\omega t}\,,
\tag{43}
$$

where $\frac{\mathrm{d}s}{\mathrm{d}e}(e_\beta) = \beta$. Thus, in the time domain Eq. (40) becomes

$$
\Gamma_{\mathrm{c}}(t) \sim \int \mathrm{d}t'\,\Gamma_{\mathrm{d}}(t-t')\mathrm{SFF}_{\mathrm{c}}(t')\,.
\tag{44}
$$

This can be interpreted as the SFF smoothed on the scale of the physical correlation function $\Gamma_{\mathrm{d}}$. Note that this is also valid at the level of the single instance, which therefore implies that the fluctuations intrinsic to the SFF also get smoothed out. At times $t \propto \mathcal{N}$, $\Gamma_{\mathrm{d}}(t-t')$ acts as a delta function $\propto \delta(t-t')$ for the slowly varying function $\mathrm{SFF}_{\mathrm{c}}$ and thus in the convolution one retrieves

$$
\Gamma_{\mathrm{c}}(t) \sim f_{e_0}^2(0)\mathrm{SFF}_{\mathrm{c}}(t)\,.
\tag{45}
$$

While the SFF and $\Gamma$ seem to behave similarly at late times, it is not the case at early times when both functions are dominated by the slope. From Eqs. (36,42), it appears that $\Gamma_{\mathrm{d}}$ decays slower than $\mathrm{SFF}_{\mathrm{d}}$ because of $f^2(\omega)$. This means that two different dip times are expected for $\Gamma(t)$ and $\mathrm{SFF}(t)$.

## 3.2 Numerical analysis in a many-body Hamiltonian

In this section, we test our predictions for two spin-glass Hamiltonians characterised by two different symmetry classes. We will focus on the infinite-temperature regime where no spin-glass phase is expected; see Ref. [38] for spectral correlations in the spin-glass phase at lower temperatures.

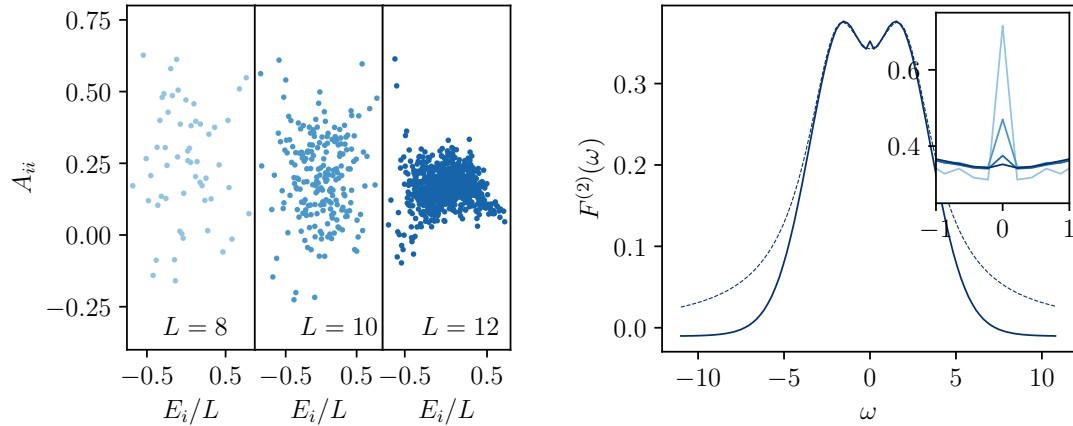

Figure 4: Left: Diagonal elements of the local observable $A$ as a function of the energy density $E_i/L$ and of the size $L$. By order of increasing sizes, the diagonal elements have a standard deviation of 0.20, 0.15, and 0.06. Right: On-shell correlations of order 2 ($F^{(2)}(\omega)$) for $L = 14$. The dashed line is a fit done with the sum of two Lorentzians. The inset shows the collapse of data for $L = 8, 10, 12, 14$ at $\omega \sim 0$.

To test our predictions for systems with GOE statistics, we consider the *XY spin glass* defined as

$$H = \sum_{i<j} J_{ij} \left( S_i^x S_j^x + S_i^y S_j^y \right), \tag{46}$$

where $J_{ij} \overset{\text{iid}}{\sim} \mathcal{N}(0, 1/\sqrt{L})$ and $\{S_i^\mu\}_\mu$ are the Pauli matrices. Assuming an even number $L$ of spins and since $[H, m] = 0$ with $m = \frac{1}{L}\sum_i S_i^z$ the magnetization, we focus on the magnetization sector $m = 2/L$ and consider the observable $A = S_{L/2}^z$. It can be numerically checked that this model does not have any degenerate levels within a magnetization sector.

### 3.2.1 GOE spin glass Hamiltonian

Before proceeding with our analysis, we wish to check the assumptions made in Sec. 3.1.

**1. The average of the observable $A$ is independent of the energy.** Fig. 4(left) shows how the diagonal elements $A_{ii} = \langle E_i | A | E_i \rangle$ of the observable vary with the energy density $E_i/L$ for sizes $L = 8, 10, 12$. As expected, the fluctuations from eigenstate to eigenstate rapidly diminish upon increasing the size $L$ and concentrate around the energy-independent average $m = 2/L$, thus verifying assumption 1.

**2. The on-shell correlations of order 2, $F^{(2)}(\omega) = f_e^2(\omega)$, have a well-defined limit when $\omega \to 0$.** As in [39], $F^{(2)}(\omega)$ can be computed for the XY spin glass by simply Fourier transforming the shifted correlation function $\Gamma(t)$ shown in Fig. 5. The result shown in Fig. 4(right) checks assumption 2. It is also worth noticing that $F^{(2)}(\omega)$ is well fitted by a sum of two shifted Lorentzians at small frequencies, which means that $\Gamma_d$ displays an exponential decay with oscillations within (as seen in Fig. 5).

The numerical results for the SFF($t$) and $\Gamma(t)$ are shown for a single Hamiltonian in Fig. 5 (right), and averaging over 1000 Hamiltonians in Fig. 5 (left). The shifted correlation function $\bar{\Gamma}(t)$ exhibits a slope-dip-ramp-plateau behaviour, just as the SFF, very visible in the left plot of Fig. 5. However, for the system sizes at our disposal, the ramp is not as linear as in the



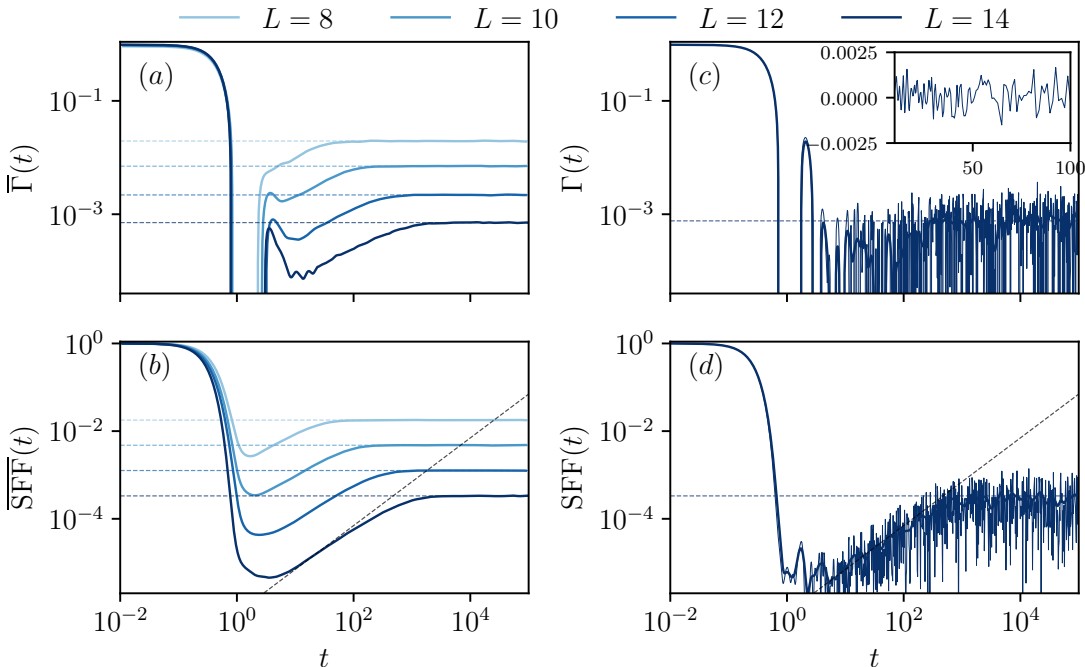

Figure 5: Left: (*a*) Shifted correlation function $\overline{\Gamma}(t)$ and (*b*) spectral form factor $\overline{\text{SFF}}(t)$ averaged over 1000 XY spin glass Hamiltonians (46) at inverse temperature $\beta = 0$. Right: (*c*) Shifted correlation function $\Gamma(t)$ and (*d*) spectral form factor $\text{SFF}(t)$ obtained from a single XY spin glass Hamiltonian (46). The inset shows some negative oscillations appearing in the ramp. Plots (*c*) and (*d*) are for $L = 14$ spins at inverse temperature $\beta = 0$. In all plots, the thin curves are the exact result while the thick curves are a smoothed version obtained by convoluting with a Gaussian kernel. The horizontal dashed lines are the expected plateau values coming from the diagonal ensemble. The increasing oblique dashed lines are fits $\propto t$.

SFF, pointing to the effect of corrections in the kernel induced by $f(\omega)$. Moreover, we note that differently from the SFF, the function $\Gamma$ is not positively defined, so in the slope, it shows negative oscillations. The fact that, instead, it remains positive at large times is a consequence of the predictions that we make about its random matrix component. Moreover, as in RMT, we clearly see that, contrarily to the SFF, the time average of $\Gamma$ does not reproduce the ensemble average in the ramp. The noise observed in Fig. 5 could be suppressed introducing some dissipation [40].

As a final remark, we add that we have checked that such a slope-dip-ramp-plateau feature is visible also in the Heisenberg spin glass $H = \sum_{i<j} J_{ij}(S_i^x S_j^x + S_i^y S_j^y)$ where $J_{ij} \overset{\text{iid}}{\sim} \mathcal{N}(0, 1/\sqrt{L})$.

### 3.2.2 GUE spin glass Hamiltonian

The generalisation of the results obtained in Sec. 2.3.2 for random matrices with GUE statistics and predicted in Sec. 3.1 for many-body systems are tested against the following *chiral spin glass*

$$H = \sum_{i<j<k} J_{ijk} \mathbf{S_i} \cdot \left( \mathbf{S_j} \times \mathbf{S_k} \right), \tag{47}$$

where $J_{ijk} \overset{\text{iid}}{\sim} \mathcal{N}(0, 1/L)$. This is a mean-field model of 3-spin chiral interactions that appear, for instance, in frustrated Hubbard models [41–44]. Notice that [42] has shown that, for

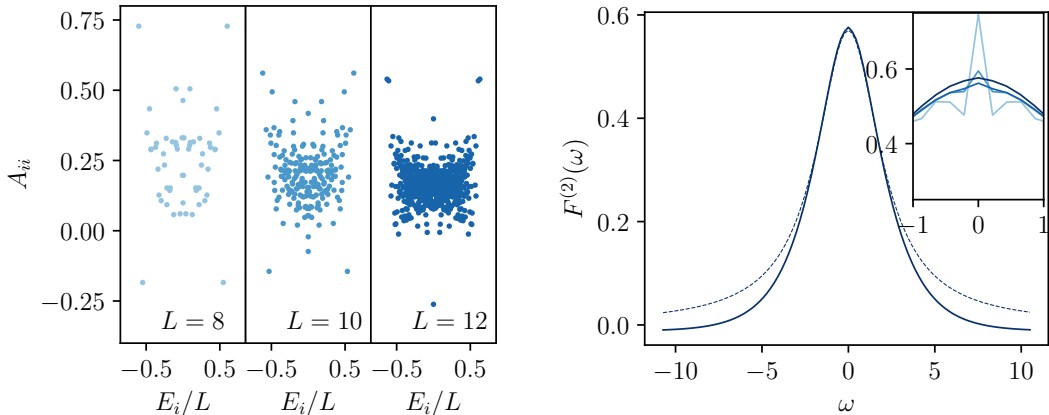

Figure 6: Left: Diagonal elements of the local observable $A$ as a function of the energy density $E_i/L$ and of the size $L$. By order of increasing sizes, the diagonal elements have a standard deviation of 0.17, 0.10 and 0.06. Right: On-shell correlations of order 2 ($F^{(2)}(\omega)$) for $L = 14$. The dashed line is a fit done with a Lorentzian. The inset shows the collapse of data for $L = 8, 10, 12, 14$ at $\omega \sim 0$.

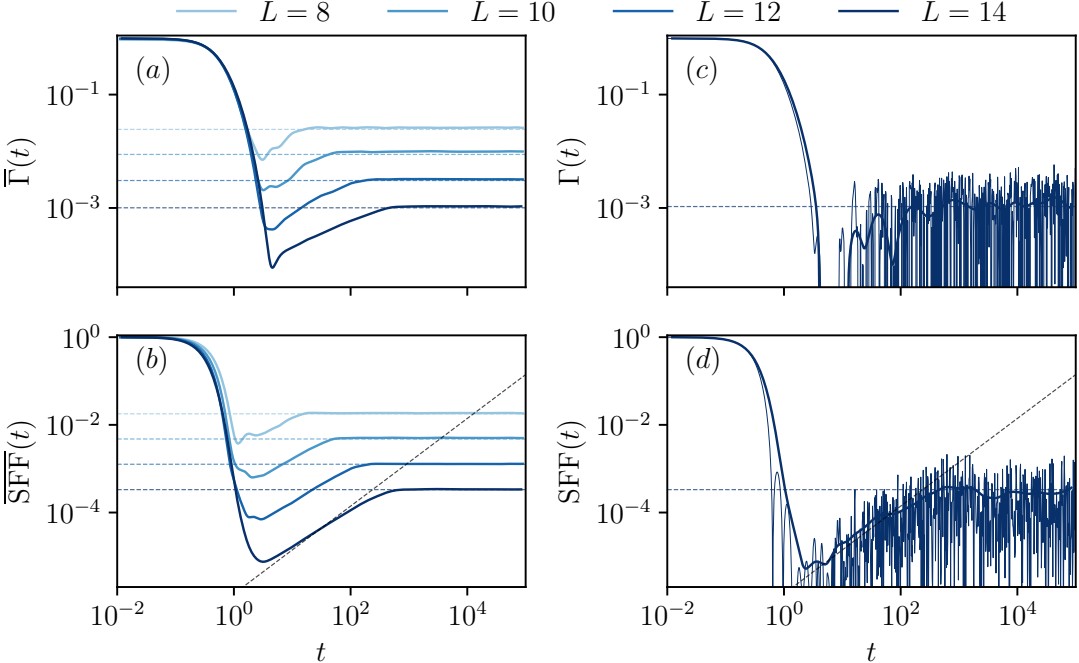

Figure 7: Left: ($a$) Shifted correlation function $\overline{\Gamma}(t)$ and ($b$) spectral form factor $\overline{\text{SFF}}(t)$ averaged over 1000 chiral spin glass Hamiltonians (47) at inverse temperature $\beta = 0$. Right: ($c$) Shifted correlation function $\Gamma(t)$ and ($d$) and spectral form factor $\text{SFF}(t)$ obtained from a single chiral spin glass Hamiltonian (47). Plots ($c$) and ($d$) are for $L = 14$ spins at inverse temperature $\beta = 0$. In all plots, the thin curves are the exact result, while the thick curves are a smoothed version obtained by convoluting with a Gaussian kernel. The horizontal dashed lines are the expected plateau values coming from the diagonal ensemble. The increasing oblique dashed lines are fits $\propto t$.

indices $i, j, k$ all different

$$\mathbf{S_i} \cdot \left( \mathbf{S_j} \times \mathbf{S_k} \right) = \frac{i}{2} \sum_{l,m,n} \varepsilon_{lmn} S_l^z S_m^+ S_n^- \,, \tag{48}$$

where $\varepsilon_{lmn}$ is the standard Levi-Civita symbol and $l, m, n$ take the values of all possible permutations of $i, j, k$. From Eq. (48), it follows that $[H, m] = 0$ with $m = \frac{1}{L} \sum_i S_i^z$ the magnetization. For an even number $L$ of spins, the analysis can, therefore, be restricted to the magnetization sector $m = 2/L$ considering the observable $S_{L/2}^z$. Numerically, it appears that this model has a few degenerate levels but, because they are so few, we can forget about them for large systems (i.e. $L$ greater than $\sim 10$). As in the previous subsection, the assumptions made in Sec. 3.1 are fulfilled as seen in Fig. 6. Moreover, the on-shell correlations $F^{(2)}(\omega)$ are well fitted by a Lorentzian at small frequency so $\Gamma_d$ has an exponential decay in time.

The SFF($t$) and $\Gamma(t)$ were computed through an exact numerical diagonalization of the Hamiltonian (47). The results are shown for a single Hamiltonian in Fig. 7 (right), and averaging over 1000 Hamiltonians in Fig. 7 (left). The slope-dip-ramp-plateau feature is well visible in this spin glass system, in both $\Gamma(t)$ and SFF($t$). Similarly to the previous case, the time average fails to reproduce the ensemble average.

### 3.2.3   Characteristic time scales

From Figs. 5 and 7, it appears that for many-body systems, both the SFF and the correlation function $\Gamma$ exhibit a similar ramp and plateau. However, the slope appearing at early times is not universal and is expected to differ for the SFF and $\Gamma$. Since the dip time $t_{\text{dip}}$ signals the crossover between the slope and the ramp, it is not the same for the SFF and for $\Gamma$ as seen in Fig. 8. On the contrary, the ramp and the plateau being two universal features, the Heisenberg time $t_{\text{Heis}}$, corresponding to the onset of the plateau, is unique.

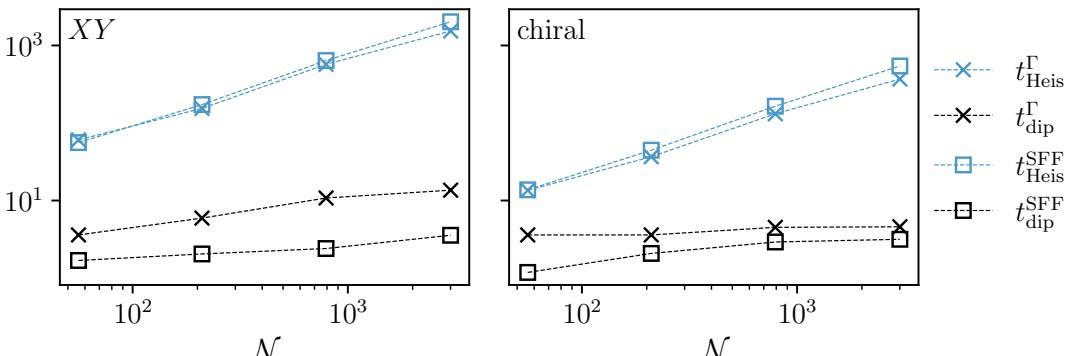

Figure 8: Dip times $t_{\text{dip}}$ and Heisenberg times $t_{\text{Heis}}$ of the SFF and the correlation function $\Gamma$ as a function of the Hilbert space dimension $\mathcal{N}$. Results for the XY spin glass (46) are in the left plot, and those for the chiral spin glass (47) are on the right. The dip times are extracted at the minimum of the SFF (or $\Gamma$) before the ramp. The Heisenberg times are defined as the first time the SFF (or $\Gamma$) reaches 95% of its plateau value given by the diagonal ensemble.

# 4 Summary and conclusions

In this work, we have performed an ETH study of dynamical connected correlation functions. As for the Spectral Form Factor (SFF), the ensemble average of such functions exhibits a ramp and a plateau at large times for well-chosen observables. This fact, already observed in the literature, has been described here using ETH arguments. In summary, our key findings are:

- In both the RMT and Hamiltonian cases, the symmetry classes are crucial for extracting eigenvalue correlations from properly shifted dynamical correlation functions, defined as $\Gamma(t) = C(t) - (1 - \beta_{\mathrm{RMT}}/2) \lim_{t\to\infty} \overline{C}(t)$, where $C(t)$ is the connected correlator. This is particularly relevant for the orthogonal ensemble ($\beta_{\mathrm{RMT}} = 1$), where the ramp is not observed without this shift.

- While the ensemble average of the SFF always coincides with a small time-window average, the same does not hold true for correlation functions during the ramp phase. This non-self-averaging behaviour of $\Gamma(t)$ emerges from the large relative fluctuations of the matrix elements of the observable under scrutiny.

- The plateau at long times is associated with the fluctuations of the diagonal matrix elements of the observables in the absence of degeneracies. To appreciate it in the Hamiltonian case, one shall consider an observable $A$ which does not overlap with the Hamiltonian; hence, the $\{A_{ii}\}_i$ fluctuate, but their averages do not depend on energy.

There are several directions for further research. For example, it would be interesting to examine the impact of locality on the early non-universal behaviour and how it influences the properties of the dip time in Hamiltonian systems. It would be valuable to explore how the transport of other hydrodynamic modes impacts these results for the observables considered here, for the same models but defined on a finite dimensional lattice.

A natural extension would be to generalise this result by studying how multi-time correlation functions encode the higher-order powers of the SFF [24]. In the context of rotationally-invariant random matrices, the powers of the spectral factors are encoded in the free cumulants [16,33]. One shall investigate its interplay with the eigenvector fluctuations in Hamiltonian systems.

Lastly, it would be interesting to identify other physical observables where eigenvalue correlations manifest in the form of a ramp. Known examples include survival probability [45,46], and this concept could be extended to the adiabatic gauge potential [47].

## Acknowledgments

We thank Gabriele di Ubaldo, Dario Poletti and Alexander Altland for useful discussions.

**Funding information**   S.P. acknowledges support by the Deutsche Forschungsgemeinschaft (DFG, German Research Foundation) under Germany's Excellence Strategy - Cluster of Excellence Matter and Light for Quantum Computing (ML4Q) EXC 2004/1 -390534769.

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
