# Peer review of "Random matrix universality in dynamical correlation functions at late times"

_SciPost Physics, doi:SciPost Phys. 19, 050 (2025)_

## Round 2 · Referee Report · Anonymous (Referee 1) · 2024-11-11

Strengths

  1. The results are clearly and concisely presented.
  2. The analytical calculations are very detailed, and extensive numerical calculations have been carried out to illustrate the ramp and plateau in correlation functions.

Weaknesses

  1. The setting that the authors focus on, namely observables whose diagonal matrix elements do not depend on energy, is very restricted. In Hamiltonian systems, these matrix elements generically depend on energy.
  2. The ramp and plateau in autocorrelation functions are exponentially small in the system size, so it is not clear that these features are observable in an experiment taking less than exponential time.

Report

Thank you for sending this manuscript. The authors study the behavior of autocorrelation functions of local observables in chaotic many-body quantum systems. They explore a late-time ramp and plateau in the time-dependence of these autocorrelation functions. Focusing on situations where the matrix elements of the observable are described by random matrix theory, and so where the time dependence is controlled by spectral statistics, it is shown that the well-known ramp and plateau in the spectral form factor imply similar features in autocorrelation functions. Understanding when such a feature should arise in autocorrelation functions is an interesting question. The ramp and plateau behavior was identified for Sachdev-Ye-Kitaev models in Ref. [7] and in chaotic Floquet systems in Ref. [25]; in the latter setting the analysis is simplified by the fact that the diagonal matrix elements of the observable do not depend on (quasi)energy.

Here, the authors also focus on the setting where diagonal matrix elements do not depend on the energy, although this situation is atypical for Hamiltonian systems. A key aim of the present work is to justify when the ramp and plateau arise, but by focusing on such a restricted setting it is not clear whether that aim has been achieved. A basic question is the degree by which diagonal matrix elements can vary while still preserving a ramp and plateau. For example, if the diagonal matrix elements vary, on average, by an amount of order unity from one end of the spectrum to the other, what happens to the ramp and plateau? A simple model in the spirit of ETH would be e.g. diagonal elements which vary linearly with energy.

The ramp and plateau in autocorrelation functions have amplitudes that are exponentially small in system size, and so it is difficult to judge their significance. Within the framework of systems described by the ETH, are there any situations where the ramp and plateau become observable (in an experiment which does not require exponential time)?

Above Eq. (30), the authors write ‘we assume for simplicity that the system is put at infinite temperature (beta=0) since using a finite temperature is a straightforward generalization’. Although it may be straightforward in the situation where the diagonal matrix elements do not depend on energy, this assumption is masking the complexity of the problem in Hamiltonian systems. It would be useful to see explicitly how e.g. Eq. (45) is modified at finite beta, at least for a simple model where the diagonal matrix elements have a simple (but nontrivial) energy dependence.

Due to the issues above, for now I will not recommend publication in SciPost. However, a revised version, which addresses when the ramp and plateau are expected (beyond the one setting of observables whose diagonal matrix elements do not depend on energy), could be suitable for publication.

Requested changes

  1. Provide additional details indicating when the ramp and plateau are expected. How are these features of autocorrelation functions suppressed if diagonal matrix elements depend on energy?
  2. Elaborate on the amplitude of the ramp and plateau, and how this amplitude depends on temperature. When are these features not exponentially small in system size?

Recommendation

Ask for major revision

---

## Round 2 · Referee Report · Anonymous (Referee 2) · 2024-11-25

Strengths

  1. It is shown (under some assumptions) that dynamical correlation functions in chaotic many-body quantum systems show a ramp and plateau structure at late times, similar to what is found in the spectral form factor.

Weaknesses

  1. As indicated in the manuscript, the connection between dynamical correlation functions and the spectral form factor has been discussed in the recent literature from other perspectives than the one given here, so the result is not completely fresh.

Report

This paper applies the eigenstate thermalisation hypothesis (ETH) to a discussion of the late-time behavior of dynamical correlation functions in chaotic many-body quantum systems. The main result is that the dynamical correlation function shows a ramp and plateau structure like the spectral form factor. More precisely, the two are related after ensemble-averaging by a (specified) linear transformation. This result is derived and tested in three ways: from random matrix theory, from ETH, and from numerics on some long-range spin glass models.

On balance, I agree with the authors' assessment that the paper: provides a novel and synergetic link between different research areas, and opens a new pathway in an existing or a new research direction. However, given prior work on the topic using approaches other than ETH, I don't think that the level of novelty or potential are as high as for some SciPost publiations.

Requested changes

  1. In addition to papers cited in the manuscript that discuss related ideas [7, 21-24, 25 and 26], the authors should cite and discuss Joshi et al, PRX 12, 011018 (2022). This paper is about the Partial Spectral Form Factor [PSFF] not correlation functions, but since the PSFF is given by an average over all correlation functions of operators with a given support, the results are very relevant to the current manuscript. While there are important differences [the PRX is about Floquet systems, while the manuscript is about Hamiltonian systems, and the PRX does not use ETH, which is the main point of the manuscript], there is also overlap in the random matrix results, especially in the idea of a shifted, connected correlation function [compare Eqns 17 and 18 of the manuscript with Eq 5 of the PRX]. Without intending to diminish the significance of the present manuscript, I think it would help readers in the field if this link were set out clearly.

  2. I think it would be useful to add more discussion of a central assumption made at the bottom of page 9 (an observable has an average that does not depend on energy density ...). For instance, it would help to give one of two examples of systems with this property at the top of page 10.

  3. In the discussion of Fig 5(c), it is asserted that ${\cal C}$ has negative oscillations but these cannot be seen in the present figure with a log axis. I think it would be useful to add an inset on a linear scale to illustrate this point.

  4. I find the notation using ${\cal C}$ and $C$ to indicate related but distinct quantities [see e.g. Eq. 17] rather confusing. I suggest that the authors should use symbols that are easier to distinguish.

  5. I don't see the reason for the superscript/power 1 on the rhs of Eqns 28 and 29. I suggest removing it or explaining the reason for this notation.

Recommendation

Publish (meets expectations and criteria for this Journal)

---

## Round 3 · Referee Report · Anonymous (Referee 2) · 2025-1-20

Report

The revised version of this paper takes account of the suggestions made in my first report, and (so far as I can see) of the suggestions made by the other referee.

I think that most of the changes are satisfactory, but I am disappointed that the reference to Joshi et al, PRX 12, 011018 (2022) that I suggested has been made in the most minimal possible way, as an extra sentence below Eq 18 and without any mention in the introduction.

Recommendation

Publish (meets expectations and criteria for this Journal)

---

## Round 3 · Referee Report · Anonymous (Referee 1) · 2025-3-3

Strengths

The paper addresses a fundamental question about autocorrelation functions in thermalizing systems. Although the feature studied here is exponentially small in the system size, the results are certainly relevant to high-precision studies of mesoscopic quantum systems.

Weaknesses

The regime where the theoretical results apply seems to be highly restricted. For now the paper focuses on the dynamics of observables whose diagonal matrix elements do not depend on energy. But, in Hamiltonian systems, and away from infinite temperatures, these matrix elements do generically depend on energy.

Report

The generic behavior in a Hamiltonian system is that there is some dependence of diagonal matrix elements on energy. Below (36), corrections are identified which are controlled by (partial_e A): how small does (partial_e A) have to be for these corrections to be negligible?

Also, since saddle point approximations are used later, do we require (partial_e A)=0 to get a ramp, or do we only require (partial_e A)=0 at the energy e0 defined by ds/de=0?

The numerical results in figures 4 and 6 are also relevant to this issue. At the beginning of 3.2.1 it is stated that ‘the average of the observable A is independent of energy’ in the GOE spin glass Hamiltonian. However, in these figures, the diagonal matrix elements have very large variations between different energy eigenstates.

There are two kinds of variations in figure 4 (left): statistical fluctuations which are expected to get smaller with increasing system size (according to the ETH) and a (more concerning) smooth dependence on energy. Based on the L=12 data in figure 4, it looks possible that a smooth dependence on energy will survive as L is increased. This raises questions over the applicability of analytical results that neglect this dependence. Similarly, on the left in figure 6, it looks like a smooth dependence of the diagonal matrix elements on energy will survive at large L.

This said, there is a linear ramp in e.g. figure 7(a). Is it possible that a ramp survives for much more general observables (than those whose diagonal matrix elements have no energy dependence) just because the quantity studied numerically is the infinite-temperature autocorrelation function? (which is dominated by energy densities with ds/de=0)

Recommendation

Ask for minor revision

---

## Round 3 · Author Response

Dear Editor,
we would like to thank both the referees for their valuable comments and appreciation of the results shown in the article. Following their suggestions and comments, we have edited certain parts of the manuscript. Below, we present a point-by-point response to all the queries of the referees.
Yours sincerely,
The authors

---

## Round 3 · List of Changes

\documentclass[11pt]{article} \usepackage{hyperref} \usepackage{graphicx} \usepackage{comment} \usepackage{xcolor} \usepackage{amsmath,amsthm,amsfonts,amssymb,amscd} \usepackage{geometry} \geometry{ left=25mm, right=25mm, top=25mm, bottom=25mm, }

\pagestyle{empty} \begin{document}

%\vspace{1cm}We thank the referees for their comments which allowed us to better express the novelty of our work.

\vspace{0.5cm} \section{Referee 1}

%\vspace{0.2cm}

\subsection{Report}

We are glad the referee appreciates the clarity of the presentation and the accuracy of the analytical and numerical results. We also thank them for the insightful comments, which we believe helped us in improving the quality of our work. Below, we address the requested changes and remarks, and we hope these revisions make the manuscript suitable for publication.\

  1. We begin by addressing concerns about the relevance of observables whose averages do not depend on energy. \ Firstly, let us give a number of instances in where such observables are found. Observables with temperature or energy-independent average appear, for instance, in systems with symmetries and conservation laws (in the manuscript, we focus on a system with a continuous symmetry and conservation of total magnetization), in disordered systems at high temperature, or in Floquet systems. (See the reply to the other referee for more details). Following the referee's remarks, we have added these comments in the resubmitted version.\ Secondly, let us remark that the literature on ETH usually focuses on energy-dependent observables. In this case, when diagonal matrix elements depend on the energy, the plateau of correlations at long times would be {\it polynomially} small in the size of the system (see, for instance, Capizzi et al. arXiv:2405.16975). This would greatly hinder the observation of the ramp, which is instead \emph{exponentially} small with the size of the system.

Nonetheless, as we mention in the paper, there has been literature pointing out in some particular case the occurrence of the ramp in correlation functions, but the previous literature discussing similar features is a bit vague on some aspects (including when to expect the occurrence of the ramp and plateau and the precise quantity to look at), so we believe it is important to state clearly all the hypotheses and the predictions.

  1. Concerning the experimental significance of the exponentially small ramp and plateau, we agree with the referee that this is a subtle effect and, for this reason, it is hard to measure it experimentally. However, let us note that these exponentially small properties that we discuss here for correlation functions, such as the ramp and plateau, are shared by the spectral form factor (SFF), which is an object widely used to diagnose chaos in quantum systems. Even if the studies on the SFF are mainly numerical experiments, they are still quite relevant in the context of many-body quantum chaos.\ We also mention that we do not hide the difficulty of observing this effect. On the contrary, we have pointed out that, unlike for the SFF, the strong fluctuations that characterize correlation functions hide the growing behavior of the ramp, making it visible only upon ensemble average. This is another feature that has not been discussed in the previous literature on the subject.

  2. Concerning the temperature-dependent generalization, we agree with the referee that the previous sentence may have been misleading. We have clarified that one needs energy-independent observables also at finite temperatures. For this reason, we referred to the generalization as ``straightforward''. We hope that the revised version is now more clear.

\subsection{Requested changes}

\begin{itemize} \item If diagonal matrix elements are dependent on the energy, the plateau would be {\it polynomially} small in the size of the system (see, for instance, Capizzi et al. arXiv:2405.16975). This would greatly hinder the observation of the ramp. We have added this comment in the main text. \item At finite temperature, in the case considered here, we would still find an exponentially small plateau (in system size). On the other hand, in the presence of energy dependencies, polynomial scaling is found. We have added to the manuscript the correction due to this effect (see Eq. 36). \end{itemize}

\section{Referee 2}

\subsection{Report}

We thank the referee for the appreciation of our work and for providing valuable feedback that has guided us in refining the manuscript. In the following, we respond to the comments and outline the changes made, which we believe address all concerns and render the manuscript ready for publication.

\subsection{Requested changes}

\vspace{0.2cm} \begin{itemize} \item We thank the referee for bringing to our attention this reference, which is definitely relevant. We have added Joshi et al, PRX 12, 011018 (2022) after Eq. 18. Note that in our case, the shift is due to the properties of the fluctuations of diagonal matrix elements, which vary in different universality classes.

\item We thank the referee for this comment, we have now added the following discussion in the manuscript: An observable that is protected by symmetry or a conservation law does not show any energy dependency. In this paper, we focus on the following mechanism. We consider a disordered spin system that conserves the total magnetization $M^z=\sum_i \sigma_i^z$ and restricts ourselves to one of the sectors of the Hilbert space with fixed total magnetization $M\in \mathbb{Z}$. In the restricted Hilbert space, the thermal average of the total magnetization is, of course, $\langle M^z\rangle_\beta=M$. Then, we consider a given site $i$ and its local magnetization $\langle \sigma_i^z\rangle_\beta$, which fluctuates depending on the disorder realization. However, upon ensemble averaging, site-permutation symmetry is restored, and $\overline{\langle \sigma^z_i \rangle_{\beta}} = M/L$. Therefore, this quantity does not depend on temperature or energy. Note that this mechanism is not restricted to spin-1/2 systems but also applies to any particle system with a $U(1)$ symmetry that implies the conservation of the total number of particles. Other examples of such observables could be drawn from disordered systems in their paramagnetic phase and at high energies where the ensemble average due to disorder implies the vanishing of expectation values of several quantities, at least in a certain range of temperatures (for instance, the (mixed) $p$-spin model in the transverse field). Similar physics has also been observed in Floquet systems and our ETH arguments can be extended to such systems. \item We have added to Fig. 5(c) an inset with a linear scale to illustrate the negative oscillations in an interval of time within the ramp. \item We have replaced the confusing notation $\mathcal{C}$ by $\Gamma$. \item The superscript/power 1 in Eqs. (27,29,31) was actually a power $-1$ but the minus sign gets mixed up with the overline in expressions such as $\overline{\rho(E)}^{-1}$ (the confusion is especially apparent when using the font used by Scipost). We have added an extra padding to fix this issue as in $\overline{\rho(E)}^{\,\,-1}$. \end{itemize}

\pagebreak

\end{document}

---

## Round 4 · Referee Report · Anonymous (Referee 1) · 2025-7-1

Report
Additionally, the forms of the corrections to correlation functions which arise in the generic case, where the diagonal part of ETH does depend on energy density, are now discussed briefly below Eq. (36), and the authors establish a contrast with [29].
In connection with this, the authors write ‘In fact, if ∂eA(e), with e = E/N the energy density, is of order one in the system’s size, there will always be polynomial corrections in L coming from the diagonal part of the matrix elements after integration by saddle-point (see below)’. Should the text read e = E/L here? Are the ‘polynomial corrections’, referenced here, corrections to the two-point function? This sentence would benefit from rephrasing.
The authors also write ‘Γ is not positively defined, so in the slope, it shows negative oscillations’, which may be an error.
The comment ‘The noise observed in Fig. 5 could be suppressed introducing some dissipation [40].’ is pretty confusing, since introducing noise or dissipation would completely change the problem. For example, the ETH would no longer be applicable.
Recommendation
Publish (meets expectations and criteria for this Journal)

Oscar Bouverot-Dupuis on 2025-05-16 [id 5486]
"3chiral_spinglass_beta1.png" plot
Attachment:
Oscar Bouverot-Dupuis on 2025-05-16 [id 5485]
"3chiral_spinglass_beta0.1.png" plot
Attachment:
Anonymous on 2025-05-16 [id 5484]
Attached to this comments (and the following) are the documents mentionned in the answer to the referees. "blue_corrected_manuscript.pdf" is the manuscript with the corrections highlighted in blue, and "2chiral_spinglass_beta0.1.pdf" and "3chiral_spinglass_beta1.pdf" are plots showing how the ramp/plateau are affected by the inverse temperature $\beta=0.1,1$ in the chiral spinglass model.
Attachment:
blue_corrected_manuscript.pdf

---

## Round 4 · List of Changes

We are glad that both Referees appreciate the quality of our work. In particular we thank Referee 1 for his/her publication recommendation and below we address the questions of Referee 2.
1 Whenever $\partial_e A(e)$, with $e=E/N$ the energy density of the saddle-point, is of order one in system's size there will always be polynomial corrections in $N$ coming from the diagonal part of the matrix elements.
1' What is important is the value at the saddle point level, as pointed out by the referee. However, note that $\partial_e A(e)$ gives the first correction, and all derivatives should be zero from our arguments. In this sense we say that the expectation value doesn't depend on the energy. Following the referee's comments, we have added a better discussion of these assumptions (highlighted in blue in the pdf attached to this answer).
2 Even if the data in Fig.4/6 may seem to depend on energy, actually these are just statistical fluctuations, disappearing as $L\to\infty$. To corroborate the data, let us note that the value of $\overline{\langle S_i \rangle} = M^z/L$ is not expected to depend on temperature (i.e. energy) in the high temperature phase of the model that we considered. In our simulation we took $M^z=2/L$ to remove eventual degeneracies at $M^z=0$.
3 We checked that the ramp persists, introducing a finite value of $\beta$, as shown in the plot attached (not shown in the manuscript). At low enough temperature the model is expected to have a spin glass transition where ETH arguments are not predicted to hold.
4 We have added that the derivation within ETH has been obtained assuming that $f(\omega=0)$ is finite.

---

## Round 5 · List of Changes

We are glad that both Referees recommend our work for publication.
Below we address the last questions of Referee 2.

\begin{enumerate}
\item[1] Yes with $e$ we mean the intensive energy $e = E/L $. The polynomial corrections we refer to concern indeed the two-point function.
We have corrected the typo in the text.

\item[2] Indeed differently from the SFF, $\Gamma$ is not positive defined so it can take negative values.
In the model we consider in the slope it shows a negative oscillation and then it goes back to be positive having a ramp and a plateau as we discuss.

\item[3] We agree that the analysis with dissipation would need a completely separate and dedicate work.
With this sentence we wanted just to give credit to some work which observed the noise is suppressed with dissipation.

---

## Editorial Decision

published